# Cell Morphology, Material Property and Ni(II) Adsorption of Microcellular Injection-Molded Polystyrene Reinforced with Graphene Nanoparticles

**DOI:** 10.3390/polym17020189

**Published:** 2025-01-14

**Authors:** Minyuan Chien, Shiachung Chen, Kuanyi Huang, Tlou Nathaniel Moja, Shyhshin Hwang

**Affiliations:** 1Department of Vehicle Engineering, Chien-hsin University of Science and Technology, Taoyuan 320678, Taiwan; jackyaren@uch.edu.tw; 2R&D Center for Smart Manufacturing, Chung Yuan Christian University, Taoyuan 32023, Taiwan; shiachun@cycu.edu.tw; 3Department of Mechanical Engineering, Chung Yuan Christian University, Taoyuan 32023, Taiwan; g11373605@cycu.edu.tw; 4Institute for Nanotechnology and Water Sustainability Research Unit, College of Science, Engineering and Technology, University of South Africa, Florida Campus, Johannesburg 1709, South Africa; 5Department of Mechanical Engineering, Chien-hsin University of Science and Technology, Taoyuan 320678, Taiwan

**Keywords:** microcellular, graphene, nanocomposites, second-order kinetic model, nickel removal, adsorption

## Abstract

Graphene’s incorporation into polymers has enabled the development of advanced polymer/graphene nanocomposites with superior properties. This study focuses on the use of a microcellular foamed polystyrene (PS)/graphene (GP) nanocomposite (3 wt%) for nickel (II) ion removal from aqueous solutions. Adsorption behavior was evaluated through FTIR, TEM, SEM, TGA, and XRD analyses. Key factors, including initial ion concentration, pH, temperature, and sorbent dosage, were examined. Results showed optimal nickel removal at specific pH levels with removal efficiency decreasing from 91 to 80% as Ni (II) concentrations increased from 10 to 100 mg/L. The adsorption capacity improved from 11 to 130 mg/g. Equilibrium data aligned with Langmuir and Freundlich isotherm models, while adsorption kinetics followed a second-order kinetic model. These findings highlight the potential of PS/GP nanocomposites for nickel ion removal, offering a promising solution for small-scale industrial applications.

## 1. Introduction

Graphene has received significant research attention since its discovery in 1994. It consists of a single layer of carbon atoms arranged in a hexagonal structure with sp^2^ hybridization. Its unique characteristics, such as high electrical and thermal conductivity, make it suitable for various engineering fields, particularly in heat dissipation, full cells, supercapacitors, and flame retardancy [1]. Graphene can be synthesized through several methods, including graphene oxide (GO), reduced graphene oxide (RGO), chemical vapor deposition, epitaxial growth [2], and the mechanical peeling method [3]. Among these, mechanical peeling is considered an environmentally friendly method as it does not involve chemicals. Microcellular foam polymer/graphene composites have diverse applications in the removal of heavy metals from wastewater due to the role of microcells. He et al. [4] employed GO and RGO to adsorb acetone gas. GO foams exhibited a saturated adsorption efficiency exceeding 100%, which was higher than that of RGO. Chen et al. [5] utilized plasma-enhanced chemical vapor deposition to fabricate three-dimensional graphene foam. This foam demonstrated high electrical conductivity, a large surface area, and an impressive absorption capacity of 177.6 mg/g for As(V) and 399.3 mg/g for Pb(II). Yang et al. [6] prepared magnetic graphene foam containing magnetite (Fe_3_O_4_) nanoparticles for adsorbing oil and organic solvents. The results indicated that graphene exhibited an excellent oil adsorption rate, making it suitable for cleaning oil spills. Jayanthi et al. [7] developed a random GO layer in a three-dimensional structure using a lyophilization technique for dye adsorption and antibacterial applications. The GO foam achieved good adsorption capabilities for carcinogenic rhodamine B (RB) dyes, malachite green, and acriflavine.

With the continuous development and rapid industrialization of clusters of mines and cities, the contamination of air [8], water, and soil [9] waste has become increasingly severe. Water pollution, in particular, significantly impacts human health and the living environment, making it a key issue that requires urgent resolution. Among global environmental concerns, water contamination is one of the most critical, particularly toxic heavy metal ions such as Pb(II), Cd(II), Ni(II), As(III), and Cr(VI) [10]. This issue poses severe environmental challenges that compromise human health worldwide [11]. Toxic heavy metal ions released from electroplating and metal-finishing plants can accumulate through the food chain, severely affecting the environment and human health. These ions are highly virulent, resistant to degradation, and widely dispersed in water [12]. Among these contaminants, Ni (II) is often labeled as a significant pollutant due to its high toxicity, mobility, and extensive use in industries such as electroplating, steel manufacturing, pigment production, and battery storage [13]. While Ni (II) plays a vital role as a micronutrient in synthesizing vitamin B12, excessive levels can lead to lung, nose, and bone cancers, as well as symptoms such as nausea, cyanosis, and rapid respiration [14]. For these reasons, the need to remove Ni (II) from industrial effluents before discharge is imperative [15]. The primary sources of nickel pollution in water include galvanization, smelting, mining, dyeing operations, battery manufacturing, and metal finishing [16]. Although trace amounts of nickel are beneficial to humans by activating certain enzyme systems, excessive levels can cause different types of diseases [17]. Various treatment methods have been developed to eliminate heavy metal ions from wastewater, including chemical precipitation, membrane separation, ion exchange, and electrochemical treatment [18]. However, each method has its advantages and disadvantages. For instance, chemical precipitation is cost-effective and easy to operate, but it produces large quantities of chemical by-products, creating landfill issues [19]. Membrane separation achieves high separation efficiency in removing heavy metal ions; however, low economic feasibility and high maintenance cost limit its large-scale application [20]. Among these methods, adsorption stands out as a fast, effective, low-cost, and relatively simple approach to wastewater treatment [21]. Various materials have been extensively utilized for adsorption, including activated carbons, smectite clay minerals, polymeric materials, modified zeolites, and mesoporous materials [22]. However, many of these materials suffer from low sorption capacity or instability at low or high pH values [23]. In recent years, researchers have noticed some threshold limitations of GP for heavy metals uptake. GP agglomerates have a high surface energy and thus are not suitable for metal ions adsorption techniques, as they eventually reduce the efficacy of the adsorption of heavy metal ions. However, graphene, a representative carbon-based nanomaterial, has emerged as a focal point for environmental researchers due to its exceptional adsorption capacity and high surface area [24]. Graphene, with its extensive surface area and unique two-dimensional structure [25], exhibits exceptional adsorption capabilities. Graphene-based materials demonstrate distinct adsorption properties [26], which are categorized as hydrophilic and hydrophobic. Hydrophilic adsorption primarily occurs through interactions between the epoxy groups on graphene oxide and hydrophilic functional groups of pollutants, significantly enhancing the removal of various toxic and hazardous contaminants from water [27]. In contrast, graphene is predominantly a “hydrophobic group”, favoring interactions with hydrophobic substances, enabling the effective adsorption of pollutants such as toxic metals [28]. Graphene’s adsorption capabilities stem from various mechanisms, including electrostatic interactions, van der Waals forces, dispersion forces, and charge transfer. Zhou et al. [29] successfully synthesized a terpyridine/graphene composite with a significant surface area of 440 m^2^/g. Similarly, Jang et al. [30] developed graphene composite membranes that demonstrated high efficiency in removing methylene blue and methyl red dyes from aqueous solutions. However, graphene-based adsorbents often suffer from limited structural stability, making them susceptible to damage during practical applications. Furthermore, graphene’s inherent tendency toward internal agglomeration can hinder its adsorption performance. Consequently, current research efforts are directed toward improving the structural integrity of graphene-based adsorbents while simultaneously mitigating the issue of internal agglomeration and the adsorption of nickel. This study incorporated polystyrene (PS) with graphene to enhance its structural integrity, ultimately improving the composite’s adsorption performance and mechanical properties. This work aims to evaluate and collect experimental data on the adsorption efficiency of graphene-polystyrene composites to remove and eliminate nickel (II) ions from aqueous solutions.

## 2. Materials and Methods

### 2.1. Material

Polystyrene (PS), PG-80, supplied by Ch-Mei Plastic Co. Ltd., Tainan, Taiwan, with a melt flow index of 11 g/10 min, was used as the matrix material in this study. PS was selected because its side chain contains styrene, which might be compatible with graphene. Graphene in nano (G2) and micro (G150) scales, with a density of 0.01–0.02 g/mL, was supplied by Knano Co. Ltd., Xiamen, China. The thin graphene (G2) nanoscale consisted of approximately three layers of single graphene sheets. Graphene was processed from graphite using a mechanical peeling method [3]. The nanographene content was varied at 0, 0.5, 1, 2, and 3 wt%. Dispersion wax was sprayed onto the surface of the PS matrix before mixing. Subsequently, nano-graphene was added to the PS matrix bag and mixed by hand shaking for two minutes.

### 2.2. Foamed Injection Molding Machine and Mold

Microcellular injection molding was performed using a 100-ton Arburg 420C (Lossberg, Germany) injection molding machine equipped with MuCell^®^ capability. Nitrogen (N_2_) served as the physical blowing agent. A supercritical fluid (SCF) booster can pump nitrogen gas up to 34 MPa, and SCF is injected into polymer melt. A shutoff nozzle is used to keep the polymer melt and scN_2_ as single phase at around 10 MPa on the barrel. Upon injection, thermal instability happens, which causes cell nucleation and cell growth during the injection process. In turn, foamed polymer is formed (Figure 1). PS/GP nanocomposites were fabricated through the following steps. The GP powder was obtained as received. Then, we sprayed dispersed wax on the PS resin and added GP powder on the PS resin. Mixing was completed by shaking the GP powder on the surface of the PS resin inside a plastic bag.

A tensile bar mold compliant with ASTM standard D-638 [31] produced the foamed polymer/graphene nanocomposites. For the heavy metal adsorption study, the tensile bar specimens were crushed into particles measuring 3~4 mm. The process conditions for conventional and microcellular injection molding are summarized in Table 1.

### 2.3. Instrumentation

The tensile test of the ASTM standard sample was conducted using a tensile test analyzer, the HT-9102M, manufactured by Hong-Da Co. Ltd., Taichung, Taiwan. For PS, the axial speed was set to 5 mm/s. Different polymers and composite samples required varying axial speeds with composites generally necessitating lower axial speeds compared to neat polymers. According to ASTM D638 standards, the tensile test should last at least 120 s. Fourier-transform infrared spectroscopy (FTIR) analysis was conducted using a Spectrum 3 Tri-Range FT-IR Spectrometer in Johannesburg, South Africa. The analysis employed scanning wavelengths ranging from 500 cm to 4000 cm^−1^. Transmission electron microscopy (TEM) was conducted using the JEM2000 by JEOL, Tokyo, Japan. Scanning electron microscope (SEM) inspection was carried out with a Sigma 360 FE-SEM, XEISS, Oberkochen, Germany.

## 3. Batch Adsorption Equilibrium

Nickel nitrate was dissolved in deionized water to prepare a 100 mg/L of Ni(II) stock solution. The effects of pH, contact time, initial concentration, and PS/GP dosage on adsorption were examined through batch adsorption experiments. A 0.6 g adsorbent was used in all experiments except for investigating the effect of pH, while the aqueous solution volume was fixed at 250 mL. Initial concentrations ranging from 10 to 100 mg/L were prepared from the 1000 mg/L stock solutions. Crushed PS/GP composites were mixed with metal ion solution, stirred using a mechanical shaker rotating at 250 rpm, and filtered through a 0.45 filter. The residual metal ion concentration was measured using an ICP-OES (Agilent 720 series). Equations (1) [32] and (2) [33] were used to calculate the amount of adsorbed metal (q_e_) and the removal efficiency (R), respectively.q_e_ = (C_o_ − C_e_)/V_m_(1)Q_max_ = (C_o_ − C_e_)/m V (2)
where q_e_ is the amount of adsorbed metal ions (mg/g), C_o_ and C_e_ are the initial and equilibrium concentrations of metal ions, respectively, V is the solution volume in liters (L), and m is the mass of the PS/GP composite in grams (g) [34].

Kinetic experiments followed the same procedure as equilibrium experiments except for the time variation. The adsorption capacity at a given time q_t_ was determined using Equation (3).q_t_ = (C_o_ − C_t_)/m V(3)
where Ct is the residual concentration (mg/L); Co is the initial concentration (mg/L); m is the mass of the adsorbent; and V is the volume of the nickel-divalent solution [35].

## 4. Results and Discussion

### 4.1. Fourier-Transform Infrared Spectroscopy (FTIR)

The fundamental peaks identified in the foamed PS/GP nanocomposite are shown in Figure 1. The characteristic of the aromatic =C–H stretching band is observed at 3024.1 cm^−1^. The symmetrical stretch of CH_2_ appeared at 2848.7 cm^−1^, while the sharp asymmetric stretching peak of –CH_2_ is observed at 2919.7 cm^−1^. The C=C para-disubstituted benzene is found at 1450.3 cm^−1^, and the aromatic skeleton is identified at 1600 cm^−1^. The O-H stretch band is attributed to water molecules available in the complex matrix at 3700 cm^−1^ wavelength. Bands between 904.9 and 695.2 cm^−1^ correspond to the vibration of C–H deformation in the aromatic ring, which is a characteristic of PS [36]. The band associated with substituted benzene in the para position (841.3 cm^−1^) confirms the aromatic ring substitution. Graphene contains functional groups on its surfaces, which likely interact significantly with the PS matrix to facilitate composite formation. This prediction is supported by the IR data. Figure 2 illustrates the key stretching frequencies of the PS/graphene composite. Similarly, a substantial shift (from 695.2 cm^−1^ to 660.7 cm^−1^) in the out-of-plane bending vibration of the PS ring in the composite indicates π–π interactions between the PS backbone and the basal planes of graphene [37]. These results demonstrate significant interactions between PS and GP, leading to the formation of a stable composite.

### 4.2. TEM

The foamed PS/GP nanocomposites were thoroughly analyzed using TEM to investigate the dispersion of GP within the PS matrix. Figure 3 shows TEM images captured at an acceleration voltage of 120 kV and a magnification of 50 kX for samples containing (a) 1 wt% GP and (b) 3 wt% GP. The micrographs demonstrate that the graphene platelets (GP) are uniformly distributed throughout the PS matrix with no significant signs of agglomeration or clustering. The well-dispersed structure suggests effective interaction between the GP and the PS polymer, which is critical for achieving enhanced material properties. The absence of visible aggregation further indicates the successful integration of GP at both 1 wt% and 3 wt% loadings, highlighting the potential of these composites for advanced applications.

### 4.3. Cell Morphology

The heavy metal adsorption capacity of the foamed nanocomposites depends on the cell size. A fixed amount of N_2_ was introduced into the polymer/nanocomposite melt. Larger cell sizes correspond to lower cell densities. Figure 4 illustrates the cell morphology of neat PS and 0.5, 1, 2, and 3 wt GP nanocomposites. For (a) neat PS and (b) 0.5 wt% GP, the cell sizes are large, and the fracture surfaces appear flat, indicating that the material is brittle. When the GP content exceeds 1 wt%, the cell size decreases, and the surface becomes rough, suggesting improved toughness compared to neat PS and 1 wt% GP composites. Figure 5 shows the relationship between the nanocomposites’ cell size and cell density. The cell sizes are 150, 124, 55, 24, and 12 μm for neat PS, 0.5, 1, 2, and 3 wt% GP, respectively. Cell densities are 2.17, 2.54, 30.0, 31.7, and 4.87 × 10^9^ Cells/cm^3^ for neat PS, 0.5, 1, 2, and 3 wt% GP, respectively. These data demonstrate that as the cell size decreases, the cell density increases [38].

### 4.4. X-Ray Diffraction Pattern

The XRD patterns of thin (G2) and thick (G150) graphenes measured from 2 to 40 degrees are presented in Figure 6. The (002) oriented diffraction peak of thick graphene (G150) at 26.56 degrees corresponds to the graphite pattern [38], which is attributed to a short crystalline range along the vertical direction. Meanwhile, thin (G2) graphene shows diffraction peaks at 18.14 and 29.49 degrees. The thick graphene exhibits sharp, intense peaks, indicating perfect crystalline formation. Different fabrication methods of graphene result in different diffraction peaks [39].

### 4.5. Tensile Strength

The tensile strength of foamed PS and the variation in GP content were examined. Figure 7 shows the tensile strength of neat PS, 0.5, 1, 2, and 3 wt% GP. The tensile strengths are 104.4, 92.4, 87.6, 104.2, and 106 MPa for neat PS, 0.5, 1, 2, and 3 wt% GP, respectively. Adding 0.5 and 1 wt% GP into the PS matrix does not improve the tensile strength due to the low GP content. A strength enhancement is observed when the GP content reaches 3 wt%. The tensile strength is also influenced by the smaller cell size, which contributes to better tensile strength (Figure 4). This suggests that nano-GP does not exhibit an aggregation problem [40]; if aggregation were to occur, the tensile strength would decrease. PS is a brittle material, and adding GP does not help elongate the foamed PS/GP nanocomposites.

### 4.6. Thermal Gravimetric (TGA) and Derivative Thermogravimetry (DTG) Analysis

Thermal gravimetric studies in Figure 8a reveal that the GP content minimally influences the thermal stability of foamed PS/graphene nanocomposites. The composite remains stable without any degradation from 100 to 400 °C. Residual material remained even after 430 °C due to carbon ash. Neat PS, with high thermal stability, shows a single-stage degradation at 420 °C. Figure 8b presents the DTG of PS/GP nanocomposites, which exhibit thermal stability similar to neat PS. The DTG values are 419.8, 418.7, 419.1, and 416.6 °C for neat PS, 1, 2, and 3 wt% GP, respectively. This can be attributed to the low graphene loading in the nanocomposite samples. Small GP loading is used to avoid aggregation problems, which may occur with nanomaterials [41]. Thus, all samples exhibit degradation characteristics similar to PS.

## 5. Kinetics Modelling

To determine the sorption capacity of neat PS and 0.5 wt% and 3 wt% GP, sorption studies were carried out by varying the concentration of metal solutions ranging from 10 to 100 mg/L at a constant room temperature, as shown in the figures below. The equilibrium adsorption isotherms for Ni(II) ions on the PS/GP composites and the kinetics parameters are described below.

### 5.1. Effect of pH and Concentration

The effect of pH on Ni(II) ion by the 3 wt% GP adsorption mechanism is found to be governed by electrostatic attraction. This was determined by varying the reaction pH of the medium from 2 to 12 by using 0.1 M NaOH and 0.1 M HCL to control pH and maintain the desired pH value. A calibrated PH-2F model pH meter was used to evaluate the pH values while keeping other parameters constant to reach the equilibration or completion of the phenomenon [42]. The surface adsorption phenomenon was observed to be highly pH-dependent. In the initial pH range of 2–4, adsorption was found to be favorable, reaching maximum capacity. As pH increases from 4 to 6, the adsorption rate slightly decreases. This trend continued (6–8), showing a decreasing trend in pH until it reached 10, where there were minimal changes in adsorption capacity, as illustrated in Figure 9 below [43]. This suggests that at lower pH, the concentration of H+ ions is low and does not compete with metal ions for complexation at the reaction sites. However, at higher pH, OH− may dominate in the reaction medium, and metal ions tend to precipitate as metal hydroxides, reducing their ability to bind with reaction sites. Hence, Figure 9 shows a higher adsorption value with increasing pH throughout the range [44]. This is because hydroxy species could strongly interact with Ni(II). The complexes’ analysis was conducted on Neat PS and 0.5 wt% GP, both of which showed a negative slope, indicating that the composite materials were not effective in the adsorption of Ni(II). Figure 9B illustrates the adsorption capacity as a function of concentration. The removal rate shows a positive slope, reaching a maximum adsorption capacity of 135 ± 0.5 mg/g at pH 2 due to less competition for binding sites and more activation sites available for Ni(II) ions [45].

### 5.2. Effect of Dosage and Time

The adsorption process depends on the amount of adsorbent, as shown in Figure 10A. The adsorption capacity increased with the increment of mass. Initially, the adsorbent was low (0.1–0.3 g) in quantity, reaching a rate of 130 ± 0.1 mg/g [46]. The adsorption rate reached equilibrium as soon as the amount of adsorbent was increased to 0.4–0.6 g. It was observed that there was no further increase in the recovery of metal ions when the amount of sorbent exceeded 0.4 g. This is because the adsorption capacity decreased beyond the optimized quantity as the reaction sites became fully occupied, leaving no more sites for further chelation or complexation. Therefore, 0.4 g was taken as the optimized amount for maximum adsorption [47].

The adsorption capacity with respect to time is shown in Figure 10B, illustrating the effect of reaction time on the removal efficiency of Ni (II) for 3 wt% GP. Figure 10B shows that the adsorption amount of Ni (II) for 3 wt% GP increased rapidly within the first 10 min. However, it levelled off during the next 10 min and reached equilibrium after 40 min. The results indicate that the adsorbent had a strong adsorption capacity for the metal ions during the first 10 min and slowly reached saturation after 40 min. The maximum adsorption capacity was 204 ± 0.2 mg/g.

### 5.3. Mechanism of Adsorption

Figure 1 below illustrates the probable bonding mechanism among PS/GP nanocomposites for the remediation of nickel-divalent metal ions. As demonstrated, GP is incorporated within PS, modifying the polymer composite with enhanced structural and chemical morphology to remediate Ni(II) ions. PS is an effective adsorbent with C=O carboxylic functional groups and is highly permeable. Graphene has a high negative charge and is an excellent sorbent for the adsorption of divalent heavy metals [48]. At pH 6, the adsorption efficiency of Cd (II) reaches equilibrium, which is likely due to an enhanced surface area and surface charge alteration of the adsorbent. As a result, the sudden decrease in efficiency of Cd (II) at pH < 7 can be attributed to the formation of Cd(OH)_2_, which induces repulsive forces between the adsorbent and adsorbate. This assumption is supported by the results in Figure 7, which shows that the optimal pH of the binary system to remediate Cd(II) from aqueous solution using 3 wt% GP is pH 6 [48].

### 5.4. Adsorption Isotherm Study

To better demonstrate the adsorption performance, the adsorption isotherm analysis of Ni(II) for 3 wt% GP was conducted [11,49]. To optimize the design of adsorption systems for Ni(II) ions removal from aqueous solutions, it is important to explain the relationship between the amount of Ni(II) ions adsorbed per unit weight of adsorbent (q_e_) at adsorption equilibrium and the residual concentration of Ni(II) ions in solution. Many empirical models are used to analyze experimental data and describe how adsorbents and adsorbents interact [50]. For example, Langmuir, Freundlich, and other isotherm models have been used to explain the results of adsorption studies.

An isotherm is a mathematical relation that provides information about the properties of an adsorbent’s interaction with a liquid based on the homogeneity/heterogeneity of the adsorbent, which can be expressed as either physical or chemical adsorption [51]. The presence and absence of competition between adsorbents for adsorption establishes a relationship between the amount of analyte adsorbed and the amount remaining in the solution at equilibrium [52]. Therefore, different concentrations of the metal ion pollutant were varied, and the adsorption of these concentrations under optimum conditions for other effective parameters was investigated using the proposed method. The desired isotherm was determined to describe the adsorption behavior based on the correlation coefficients of the linear forms of the conventional isotherm equations. Table 1 shows the values of isotherm correlation coefficients based on the linear equations. Two classical adsorption equilibrium models, namely Langmuir (Equation (4)) and Freundlich equation (Equation (6)), have been used in several studies on metal ions sorption by polymer composites [53].

#### 5.4.1. Langmuir Isotherm

Whenever the plot fits a Langmuir isotherm, a plot of Ce/qe versus Ce is linear, as shown in Figure 8. The R^2^ values (R^2^ = 0.235) indicate that the adsorption data for 3 wt% GP does not follow the Langmuir model. The functional groups of CH, OH-, and C=O have been applied to uptake Ni(II) from aqueous solutions in the 3 wt% GP polymer composite. The results do not conform to the Langmuir adsorption model. A crucial component of the Langmuir adsorption model is the equilibrium factor RL. RL is a dimensionless constant used to determine whether the adsorption system is favorable [54]. RL is calculated from the initial concentration using Equation (4). If RL ranges between 0 and 1, the sorption is considered favorable, indicating that sorption occurs at specific homogeneous sites on the adsorbent [55]. The dimensionless constant was found to be 0.99, suggesting favorable sorption.q_e_ = q_m_ K C_e_/(1 + K C_e_) (4)
where q_e_ is the adsorption capacity at each concentration (mg/g); q_m_ is the maximum adsorption capacity or the adsorption capacity when the system achieves equilibrium (mg/g); K is the equilibrium constant (mg/L); and C_e_ is the adsorbate concentration (mg/L).RL = 1/(1 + K C_o_) (5)
where K is related to the energy of adsorption (l/mg), and C_o_ is the initial concentration.

#### 5.4.2. Freundlich Isotherm

The Freundlich isotherm model is based on a correlation between the adsorption of solutes from a liquid to a solid surface and assumes that the adsorbent surface is heterogeneous, involving multiple sites with varying adsorption energies. The nonlinear form of Freundlich isotherm is given by Amrutha et al. as [56](6)$$q_{e}=K_{f}C_{e}^{1/n}$$
where q_e_ is the adsorption capacity for each concentration (mg/g); K_f_ is the Freundlich constant (Lm/g); C_e_ is the adsorbate concentration (mg/L); and 1/n is the factor for heterogeneity.

The strong linear correlation observed in the plots, with coefficients exceeding 0.9, indicates that the adsorption process closely follows the Freundlich isotherm model. As evidenced in Figure 11, a correlation coefficient of 0.95 was obtained, confirming the high effectiveness of the 3 wt% GP composites for Ni(II) uptake, demonstrating an adsorption capacity of 204.4 mg/g as shown in Table 2.

#### 5.4.3. Temkin Isotherm


(7)
$$Q_{e}=\frac{RT}{B_{T}}\ln K_{T}+\frac{RT}{B_{T}}lnC_{e}$$


Expressed by Equation (7), the Temkin isotherm equation incorporates parameters such as the universal gas constant (R), the Temkin isotherm constant related to the heat of adsorption (B_T_), the Temkin isotherm constant (K_T_), and absolute temperature (T). The Temkin constant (B_T_) is calculated as 72.86 J·mol^−1^, emphasizing the prevalence of chemisorption on a homogeneous surface. The equilibrium binding constant (K_T_) is determined as 1.16 L·g^−1^, with a low correlation coefficient (R^2^ < 0.10) across all adsorbates, suggesting that the calculated values does not satisfy the experimental data of the model.

The Temkin isotherm model is particularly relevant for systems with heterogeneous surface energy, where the heat of adsorption decreases linearly with coverage due to interactions between the adsorbate and adsorbent. This model aids in understanding the adsorption process and the variation in sorption heat with increasing coverage, providing valuable insights into the binding energy dynamics.

The linear form of the Lagergren pseudo-first-order kinetic model can be expressed aslog(q_e_ − q_t_) = log q_e_ − Kad/2.303 t (8)

Here, q_e_ and q_t_ represent the amounts of Ni(II) adsorbed onto the 3 wt% GP polymer composites at equilibrium and at a specific time t (minutes), respectively, while Kad denotes the pseudo-first-order rate constant (min^−1^). The values of Kad and the correlation coefficients for varying Ni(II) ion concentrations were determined from the linear plots of log(q_e_
*−* q_t_) against t. A best-fit line was generated, and the corresponding correlation coefficient (R^2^) was calculated, as illustrated in Figure 12A. However, the low correlation coefficient suggests that the adsorption of Ni(II) ions onto the polymer composites does not conform to pseudo-first-order kinetics.

The pseudo-second-order kinetic model is represented by Equation (9). For this model to be applicable, a linear relationship must exist between t/q(t) and t. Figure 12B illustrates this linear trend with a correlation coefficient (R2) of 0.93. This confirms that the adsorption of Ni(II) ions follows the pseudo-second-order model, as summarized in Table 3. A correlation coefficient exceeding 0.9 is typically considered necessary to validate the second-order kinetic model. Therefore, the sorption process of Ni(II) ions onto the 3 wt% GP polymer composites aligns with the pseudo-second-order kinetics.

The linear form of the Lagergren et al. [57] Equation (9) for the pseudo-second-order is expressed as(9)$$\frac{t}{q_{t}}=\frac{1}{K_{2}q_{e}^{2}}+\frac{1}{q_{e}}$$

To apply the pseudo-second-order kinetic model, the plot of t/q(t) versus t must exhibit a linear trend. Figure 12B presents this plot, which shows a strong linear fit with a correlation coefficient (R^2^) of 0.98 for Ni(II), indicating the suitability of the pseudo-second-order model. The correlation coefficients for this model exceeded 0.9, confirming the second-order nature of the adsorption process for Ni(II) on the PS/GP 3 wt% polymer composite. Additionally, the Freundlich isotherm was found to better describe the experimental data compared to the Langmuir model for Ni(II) ion removal using the PS/GP 3 wt% polymer composite, as evidenced by the higher correlation coefficient (R^2^).

### 5.5. Desorption Studies

Figure 13 depicts the desorption studies of Ni(II) using PS/GP 3 wt% nanocomposites in aqueous solutions. Desorption is essential for facilitating the reuse and recycling of the adsorbent across multiple cycles. Consequently, the repeated adsorption–desorption cycles after regenerating the adsorbent serve as critical indicators of its efficiency and sustainability. The adsorption test was conducted at 25 °C, pH 2.5 ± 0.5, an initial Ni(II) concentration of 100 mg/L, a 0.6 g adsorbent mass, 250 mL solution volume, and a 40-minute contact time. The desorption experiment was conducted at 25 C, pH 9.5, using 1.0 M NaOH desorption solutions with a 100 ppm initial Ni(II) concentration and a 40-minute contact time. The desorption removal rate of Ni(II) concentration from PS/GP 3 wt% decreases significantly with an increasing recycling cycles. This led to early equilibrium, which resulted in an increased residual concentration of Ni(II) ions in the solution as the cycles progressed. This trend is evident in Figure 13, where the removal rate decreased from 93% to 44% after ten cycles. This suggests that the adsorbent remains efficient and can be recycled for up to ten cycles.

## 6. Conclusions

Instrumental analysis of the organic nanocomposite demonstrates a successful synthetic route for fabricating foamed PS/GP polymer nanocomposites with varying weight percentages, ranging from 0.5 to 3 wt%. The increased weight percentage of the PS/GP polymer composite led to an enhancement in the strength of the polymer nanocomposite, as observed in Figure 6. The increase in cell density is attributed to graphene loading in the polystyrene. Functional groups were identified using FTIR-ATR, and all samples exhibited characteristics of the parent material, which is attributable to the low graphene loading. Nano-graphene did not significantly affect the thermal stability. The prepared PS/GP polymer nanocomposites, with ratios of 0.5, 1, and 3 wt% GP, were used as heavy metal adsorbents for the adsorption of Ni(II) from aqueous solutions. The removal of Ni(II) on PP/GP polymer nanocomposite followed the Langmuir isotherm, indicating that Ni(II) ions were adsorbed on homogeneous sites with uniform binding energies, forming a layer on the nanocomposite surface. It was established that removing Ni(II) from aqueous solutions occurred predominantly through chemical adsorption, which was consistent with pseudo-second-order kinetics.

## Data Availability

The original contributions presented in this study are included in the article. Further inquiries can be directed to the corresponding authors.

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
