# Peer review of "Cell Morphology, Material Property and Ni(II) Adsorption of Microcellular Injection-Molded Polystyrene Reinforced with Graphene Nanoparticles"

_polymers, 2025, doi:10.3390/polym17020189_

Round 1
Reviewer 1 Report
Comments and Suggestions for Authors
In the manuscript of Polymers-3426289, the authors described a promising work, revealing a good methods for preparing Polystyrene reinforced graphene nanoparticles. However, its current form did not meet the requirement, I suggest a major revision.
(1) In the Introduction, some contents are too general. For example, the source/methods for GO/RGO is unnecessary. I suggest that the emphsize should be given on the disadvantages of directly using GO in environmental application to state why current GO need the reinforcement.
(2) Line 58-113, why to choose Ni as a target pollutants, as other ions mentioned in this work show more risk on the environment and health. Moreover, the reason of choosing Ni should consider the specific structural characteristics of PS/PG. Otherwise, the innovation of this work is not clear.
(3) Some information about microcellular GO should be provided, if it could be simply synthesized from Peeling? What is the key step to form the microcelluar structure.
(4) data regading the adscorption performance should be given as mean +- std.
(5) Diagram 1, suggested to add the information about PS in GP mateiral.
(6) Line 268, it should be Figure 8. Need to check the entire manuscript.
(7) After 10 cycles of test, the material lost almost half performance, what is the reason?
Hope these suggestions could help the author inproving their work.
Author Response
Dear Reviewers: Thank you for your insight comments and suggestions. My answers are marked in red are as follows.
Suggestions/Correction:
Reviewer 1:
- In the Introduction, some contents are too general. For example, the source/methods for GO/RGO is unnecessary. I suggest that the emphasize should be given on the disadvantages of directly using GO in environmental application to state why current GO need the reinforcement
Answer: Line 88, the disadvantages of using GO is mentioned and illustrated. “In recent years researchers have noticed some threshold limitations of GP for heavy metals uptake. GP turn to agglomerate owing to have high surface energy and are not suitable for metal ions adsorption technique and eventually reduces the efficacy in adsorption of heavy metal ions.”
- Line 58-113, why to choose Ni as a target pollutant, as other ions mentioned in this work show more risk on the environment and health. Moreover, the reason of choosing Ni should consider the specific structural characteristics of PS/PG. Otherwise, the innovation of this work is not clear
Answer: Almost all heavy metals are toxic when consumed at high or large quantity over time. As mentioned in line 67 “Among these contaminants, Ni (II) is often labelled as a significant pollutant due to its high toxicity, mobility, and extensive use in industries such as electroplating, steel manufacturing, pigment production, and battery storage [13]”. Ni(II) is also carcinogenic and pose harmful to human health if consumed at high quantity.
Line “109”: and adsorption of nickel
- Some information about microcellular GO should be provided, if it could be simply synthesized from Peeling? What is the key step to form the microcellular structure.
Answer: A supercritical fluid (SCF) booster can pump Nitrogen gas up to 35 MPa and SCF is in-jected into polymer melt. A shutoff nozzle is used to keep the polymer melt and scN2 as single phase at around 10 MPa on the barrel. Upon injection, thermal instability happens which causes cell nucleation and cell growth during injection process. In Turn, foamed polymer is formed (Fig. 1). PS/GP nanocomposites was fabricated through the following steps. The GP powder was obtained as received. Then sprayed dispersed wax on the PS resin and add GP powder on the PS resin. Mixing was done by shaking the GP powder on the surface of PS resin inside a plastic bag.
Figure 1: Microcellular injection molding setup.
- Data regarding the adsorption performance should be given as mean +- std.
Answer: line 315, 324, 342 values has been presented with the format of mean ±std
- (5) Diagram 1, suggested to add the information about PS in GP mateiral.
Answer: Thank you very much for noticing that, the corrected diagram has been inserted and changed to the one below.
- (6) Line 268, it should be Figure 8. Need to check the entire manuscript.
Answer: All Figure’s numbers are double checked and corrected.
- (7) After 10 cycles of test, the material lost almost half performance, what is the reason?
Answer: Desorption depends on factors such as the strength of bonding to the surface and functional groups available in the composite. After the 10th cycle the efficiency is lower due to loss of active cites and deformation of the structure. From the beginning of desorption study there is slight decrease in the efficiency uptake of Nickel ions. As observed from figure 12, after the 7th cycle, the adsorbent efficiency remained the same, indicating equilibrium in ions uptake. Illustrating that the composite cannot adsorb any further.
Reviewer 2 Report
Comments and Suggestions for Authors
Overall, the work is well-structured, and this contribution should be considered for publication after addressing the following comments.
1. Re-write and organize the abstract, The abstract should be clear and concise, providing a brief overview of the study's purpose, methods, and findings. Explain that The adsorption characteristics of PP/GP composites (wt%) were analyzed using FTIR for chemical bonding, TEM and SEM for morphological studies, TGA for thermal stability, XRD for crystallinity, DMA for mechanical properties, etc. Moreover, the keywords should be more specific.
2. Equations 1 and 2 need references.
3. Figure 1 indicates the O-H peak; however, the text does not discuss it for C=O.
4. Section 4.2 on TEM requires further discussion to provide more comprehensive insights and the figure legend needs additional details for clarity.
5. In section 5.1, This was determined by varying the reaction pH of the medium from 2 to 12 (how the pH was maintained, write in detail.
6. Did the authors study other models, such as the Temkin isotherm or similar, to further analyze the adsorption or thermal behavior of the composites? If not, could this approach provide additional insights into the material properties?
Author Response
Dear Reviewers: Thank you for your insight comments and suggestions. My answers are marked in red are as follows.
Suggestions/Correction:
Reviewer 2
- Re-write and organize the abstract, The abstract should be clear and concise, providing a brief overview of the study's purpose, methods, and findings. Explain that The adsorption characteristics of PP/GP composites (wt%) were analyzed using FTIR for chemical bonding, TEM and SEM for morphological studies, TGA for thermal stability, XRD for crystallinity, DMA for mechanical properties, etc. Moreover, the keywords should be more specific.
Answer: Abstract is revised as: Graphene's incorporation into polymers has enabled the development of advanced poly-mer/graphene nanocomposites with superior properties. This study focuses on the use of a microcellular foamed polystyrene (PS)/graphene (GP) nanocomposite (3 wt%) for nickel (II) ion removal from aqueous solutions. Adsorption behavior was evaluated through FTIR, TEM, SEM, TGA, and XRD analyses. Key factors, including initial ion concentration, pH, temperature, and sorbent dosage, were examined. Results showed optimal nickel removal at specific pH levels, with removal efficiency decreasing from 91 to 80% as Ni (II) concen-trations increased from 10 to 100 mg/L. The adsorption capacity improved from 11 to 130 mg/g. Equilibrium data aligned with Langmuir and Freundlich isotherm models, while adsorption kinetics followed a second-order kinetic model. These findings highlight the potential of PS/GP nanocomposites for nickel ion removal, offering a promising solution for small-scale industrial applications.
Keywords: microcellular, graphene, nanocomposites, second-order kinetic model, nickel removal, adsorption’
- Equations 1 and 2 need references.
Answer: Thank you for the suggestion. Line 172. Equation 1 [31]
[31] Murphy, O.; Vashishtha, M.; Palanisamy, P.,; Kumar, K.; A review on the adsorption isotherms and design calculations for the optimization of adsorbent mass and contact time. American Chemical Society, 2003, 8, 20, 17403-17430.
Line 173 Equation 2 [32]
[32]: Saba, A.; Langmuir, Freundlich and Temkin adsorption isotherms and kinetics or the removal aartichoke tournefortii straw from agricultural. Journal of Physics, 2020,1664, 1-10.
- Figure 1 indicates the O-H peak; however, the text does not discuss it for C=O.
Answer: Thank you for your suggestions and corrections. The corrected FTIR spectra has been incorporated. In the composition of GPPS there is no C=O peak, However, (Line 180) the O-H can be attributed to water molecules available in the complex matrix at 3700 cm-1wavelength.
- Section 4.2 on TEM requires further discussion to provide more comprehensive insights and the figure legend needs additional details for clarity.
Answer: The foamed PS/GP nanocomposites were thoroughly analyzed using TEM to investigate the dispersion of GP within the PS matrix. Figure 3 shows TEM images captured at an acceleration voltage of 120 kV and a magnification of 50 kX for samples containing (a) 1 wt% GP and (b) 3 wt% GP. The micrographs demonstrate that the graphene platelets (GP) are uniformly distributed throughout the PS matrix, with no significant signs of agglomeration or clustering. The well-dispersed structure suggests effective interaction between the GP and the PS polymer, which is critical for achieving enhanced material properties. The absence of visible aggregation further indicates the successful integration of GP at both 1 wt% and 3 wt% loadings, highlighting the potential of these composites for advanced applications.
- In section 5.1, This was determined by varying the reaction pH of the medium from 2 to 12 (how the pH was maintained, write in detail.
Answer: (Line 273) This was determined by varying the reaction pH of the medium from 2 to 12 by using 0.1M NaOH and 0.1M HCL to control pH and maintain the desired pH value. A calibrated PH-2F model pH meter was used to evaluate the pH values.
- Did the authors study other models, such as the Temkin isotherm or similar, to further analyze the adsorption or thermal behavior of the composites? If not, could this approach provide additional insights into the material properties?
Answer: Yes, thank you for that input, it is valuable. Line 414: and Temkin. The authors included Temkin isotherm
(6)
Expressed by Equation 6, the Temkin isotherm equation incorporates parameters such as the universal gas constant (R), the Temkin isotherm constant related to the heat of adsorption (BT), the Temkin isotherm constant (KT), and absolute temperature (T). The Temkin constant (BT) is calculated as 72.86 J·mol−1, emphasizing the prevalence of chemisorption on a homogeneous surface. The equilibrium binding constant (AT) is determined as 1.155 L·g−1, with a low correlation coefficient (R2 < 0.10) across all adsorbates, suggesting a poor t of experimental data to the model.
The Temkin isotherm model is particularly relevant for systems with heterogeneous surface energy, where the heat of adsorption decreases linearly with coverage due to interactions between the adsorbate and adsorbent. This model aids in understanding the adsorption process and the variation in sorption heat with increasing coverage, providing valuable insights into the binding energy dynamics.
Round 2
Reviewer 1 Report
Comments and Suggestions for Authors
My comments are well addressed. The manuscript could be considered to be accepted as its current form.
Reviewer 2 Report
Comments and Suggestions for Authors
The author successfully addressed all the comments and accepted the paper.